# Differentiation of Adipose-Derived Stem Cells into Smooth Muscle Cells in an Internal Anal Sphincter-Targeting Anal Incontinence Rat Model

**DOI:** 10.3390/jcm12041632

**Published:** 2023-02-17

**Authors:** Minsung Kim, Bo-Young Oh, Ji-Seon Lee, Dogeon Yoon, You-Rin Kim, Wook Chun, Jong Wan Kim, Il Tae Son

**Affiliations:** 1Department of Surgery, Hallym Sacred Heart Hospital, College of Medicine, Hallym University, Anyang 14068, Republic of Korea; 2Burn Institute, Hangang Sacred Heart Hospital, College of Medicine, Hallym University, Seoul 07247, Republic of Korea; 3Department of Surgery, Hangang Sacred Heart Hospital, College of Medicine, Hallym University, Seoul 07247, Republic of Korea; 4Department of Surgery, Dontan Sacred Heart Hospital, College of Medicine, Hallym University, Hwaseong-si 18450, Republic of Korea; 5Institute for Regenerative Medicine, Hallym Sacred Heart Hospital, College of Medicine, Hallym University, Anyang 14068, Republic of Korea

**Keywords:** adipose stem cells, smooth muscle cells, internal anal sphincter, fecal incontinence

## Abstract

Objective: Studies on development of an anal incontinence (AI) model targeting smooth muscle cells (SMCs) of the internal anal sphincter (IAS) have not been reported. The differentiation of implanted human adipose-derived stem cells (hADScs) into SMCs in an IAS-targeting AI model has also not been demonstrated. We aimed to develop an IAS-targeting AI animal model and to determine the differentiation of hADScs into SMCs in an established model. Materials and Methods: The IAS-targeting AI model was developed by inducing cryoinjury at the inner side of the muscular layer via posterior intersphincteric dissection in Sprague–Dawley rats. Dil-stained hADScs were implanted at the IAS injury site. Multiple markers for SMCs were used to confirm molecular changes before and after cell implantation. Analyses were performed using H&E, immunofluorescence, Masson’s trichrome staining, and quantitative RT–PCR. Results: Impaired smooth muscle layers accompanying other intact layers were identified in the cryoinjury group. Specific SMC markers, including SM22α, calponin, caldesmon, SMMHC, smoothelin, and SDF-1 were significantly decreased in the cryoinjured group compared with levels in the control group. However, CoL1A1 was increased significantly in the cryoinjured group. In the hADSc-treated group, higher levels of SMMHC, smoothelin, SM22α, and α-SMA were observed at two weeks after implantation than at one week after implantation. Cell tracking revealed that Dil-stained cells were located at the site of augmented SMCs. Conclusions: This study first demonstrated that implanted hADSc restored impaired SMCs at the injury site, showing stem cell fate corresponding to the established IAS-specific AI model.

## 1. Introduction

Regenerative medicine using cells has evolved as an optional treatment for anal incontinence (AI) [1]. According to previous systematic reviews of AI cell therapy, most experimental studies have been performed using animal models associated with injury or damage to the external anal sphincter (EAS) or a nerve [2,3].

To restore the damaged skeletal muscle of the EAS, several preclinical or human pilot studies have reported efficacy and safety, showing accumulated data using muscle-derived cells, autologous skeletal-muscle-derived cells, autologous myoblasts, muscle progenitor cells, and autologous myogenic satellite cells [4,5,6,7,8,9,10,11,12]. However, there is a lack of data indicating significant improvement in resting pressure in anorectal manometric measurements associated with smooth muscle cells (SMCs) of the internal anal sphincter (IAS) [4,5,6,7,12]. Only Frudinger et al. [4,5] have shown increased resting pressure, without establishing an association with injected autologous skeletal muscle-derived cell injection into SMCs at the cellular level. The current status of the regenerative approach using SMCs for the IAS is associated with a low specificity based on the studies involved [2,3]. The histologic and myogenic functions of SMCs in an IAS differ from those of skeletal muscle cells in an EAS [13]. However, the experimental study of specific incontinence models targeting SMCs in the IAS requires further clarification by reducing the bias caused by skeletal muscle cells in the EAS [14].

Among several cellular therapies, human adipose-derived stem cells (hADScs) have been applied to repair damaged or dysfunctional tissues [15,16]. Furthermore, with their easy isolation method, wide availability, and safer access, studies using these cells have been widely employed in animal and human research [17,18]. However, there have been debates regarding the therapeutic mechanism underlying implanted hADScs in cell therapy for AI. In those studies [18], injured sphincter muscle fibers were healed via granulation of fibrous tissue with infiltrated muscle fibers and not by newly differentiated muscle fibers of injected stem cells [19]. Experimental evidence indicating the differentiation of hADScs in AI cell treatment has not been reported [20,21,22,23]. Furthermore, the differentiation of implanted hADScs into corresponding SMCs in an IAS-targeting incontinence model has not been demonstrated.

The restoration of dysfunctional smooth muscle in the IAS, which provides more than 70% of the pressure necessary for the maintenance of normal continence, is critical for AI patients with decreased resting pressure, although the EAS and hemorrhoidal tissue contribute to maintenance of the resting pressure [24]. In this study, we aimed to develop an IAS-targeting incontinence animal model and determine the differentiation of hADScs into SMCs in the established IAS-targeting animal model.

## 2. Materials and Methods

### 2.1. In Vitro Differentiation of hADScs into SMCs

The hADScs provided by Anterogen Co., Ltd. (Seoul, Republic of Korea) for use in previous studies [17,18] were cultured in minimum essential medium (MEM) Alpha+ GlutaMAX (Thermo Fisher Scientific, Waltham, MA, USA) containing 0.1% gentamicin (Thermo Fisher Scientific), MycoZap Plus-PR (Lonza, Basel, Switzerland), and 10% fetal bovine serum (FBS; Thermo Fisher Scientific). For efficient differentiation into SMCs, hADScs (1.5 × 10^5^ cells/well) were seeded in fibronectin (FN)-coated 60 mm culture dishes and incubated in an expansion medium at 37 °C in a humidified 5% CO_2_ atmosphere. The next day, the medium was exchanged for smooth muscle inductive medium (Medium MCDB 131 supplemented with 1% FBS plus 100 units/mL heparin or low-glucose Dulbecco’s modified eagle medium (DMEM; Thermo Fisher Scientific) containing 0.1% gentamicin plus 5 ng/mL transforming growth factor β (TGF-β)) for smooth muscle differentiation. Thereafter, the medium was changed every 2~3 days. The cells were harvested after two weeks to confirm whether the cells had differentiated into SMCs.

### 2.2. RNA Isolation and Quantitative Real-Time Polymerase Chain Reaction (RT–PCR)

Total RNA was isolated using Easy-Blue reagent (Intron Biotechnology, Sungnam, Republic of Korea) according to the manufacturer’s protocol. Briefly, Easy-Blue reagent was removed by adding chloroform, and mRNA was precipitated using isopropanol. The RNA precipitates were washed with 75% ethanol. The quantity and purity of RNA were assessed via optical density measurements at 260 nm and 280 nm using an ultraviolet spectrometer, and the integrity of the RNA was confirmed via agarose gel electrophoresis. Total RNA (500 ng) was reverse-transcribed into 10 μL cDNA using 5× PrimeScriptTM RT Master Mix (Perfect Real Time, cat. #RR036A, Takara, Kusatsu, Japan) under the following conditions: reverse transcription at 37 °C for 15 min, heat inactivation of reverse transcriptase at 85 °C for 5 s, followed by a final hold at 4 °C. Thereafter, cDNA was diluted 10-fold with nuclease-free water. Gene-specific primers are listed in Table 1. Reactions were performed in a mixture (20 μL) containing 4 μL of 5 μM gene-specific primers, 6 μL template cDNA, and 10 μL 2× SYBR (TB Green^®^ Premix Ex Taq™, cat. #RR420L, Takara, Kusatsu, Japan) under the following conditions: denaturation at 95 °C for 5 min; 40 cycles of 95 °C for 15 s, and 60 °C for 34 s, with a final extension at 72 °C for 5 min. Reactions were performed using a Roche LC96 instrument (Roche Diagnostics, Penzberg, Germany). PCR was performed to validate the results, and the products were separated on a 2% agarose gel and visualized using red safe.

### 2.3. Immunofluorescence Staining

Cells grown or differentiated on round glass coverslips in 24-well plates were fixed and permeabilized with 100% cold methanol for 10 min. Fixed cells were incubated for 1 h in phosphate-buffered saline with Tween 20 (PBST; PBS + 0.1% Tween 20) containing 3% bovine serum albumin for blocking, followed by incubation for 2 h at 37 °C with specific primary antibodies. Anti-calponin 1 antibody (ab46794) was obtained from Abcam (Cambridge, UK) and anti-alpha-smooth muscle actin (α-SMA; a5228) antibody was purchased from Sigma-Aldrich (St. Louis, MO, USA). Cells were washed three times with PBST and incubated with Cy2-conjugated goat anti-rabbit/mouse IgGs (Jackson Immunoresearch Laboratories, West Grove, PA, USA) or Alexa 594-conjugated goat anti-rabbit/mouse IgGs (Molecular Probes, Eugene, OR, USA) as required. Cellular DNA was counterstained with 4′,6-diamidino-2-phenylindole (DAPI; 0.2 μg/mL in PBS).

### 2.4. Histologic Staiing

Each tissue sample was fixed with 4% paraformaldehyde, paraffin-embedded, and sectioned into 3 μm-thick sections. For hematoxylin and eosin (H&E) staining, the slides were deparaffinized, rehydrated, and stained with H&E. Masson’s trichrome staining (connective tissue stain) was performed according to the manufacturer’s instructions (#SS1026-MAB-500, CANCER). Briefly, cryosection slides were placed in preheated Bouin’s fluid for 60 min, followed by a 10 min cooling period. The slides were rinsed in tap water until the sections were completely clear and then washed once in distilled water. The slides were then stained with equal volumes of Weigert’s A and B for 5 min and rinsed with running tap water for 2 min. Next, the slides were exposed to Biebrich scarlet-acid fuchsin solution for 15 min and rinsed with distilled water. The slides were differentiated in phosphomolybdic/phosphotungstic acid solution until collagen was no longer red and then rinsed with distilled water. Without further rinsing, the slides were treated with aniline blue solution for 5–10 min, followed by treatment with 1% acetic acid for 3–5 min and rapid dehydration with two changes of 95% and 100% ethanol. Finally, the slides were incubated with xylene and mounted with balsam.

### 2.5. IAS-Specific Incontinence Model Development and hADSc Implantation

Sprague–Dawley rats (female, 250–300 g, 7 weeks old) were housed in the local animal care facility according to institutional guidelines. All procedures were performed under anesthetic at an appropriate depth using isoflurane anesthesia with an initial dose of 3% and a maintenance dose of 2.5%. Ten rats were used for the IAS-cryoinjured incontinence animal model. Rats were randomly assigned to the cryoinjury group (*n* = 5) and the normal group (*n* = 5). An initial incision (2–3 mm) was made transversely at the posterior intersphincteric groove to preserve the muscular layer of the EAS. Non-thermal caudal to cranial dissection was performed through the intersphincteric space. Using a 25-gauge needle chilled with liquid nitrogen, cryoinjury was inflicted for less than 30 s at the ventral muscular layer connected between the submucosal and muscular layers. The normal group did not undergo any procedure on the anal sphincter. After dissection, the sphincter wound was repaired with interrupted sutures using 5-nylene (Figure 1). The IAS specimen was harvested near the anal verge via removal from the distal rectum 1 week after cryoinjury, while considering the different myogenic features of SMCs in the IAS from those in the rectum [2,25]. After harvesting, all experimental animals were euthanized using carbon dioxide gas.

After confirming the *in vitro* differentiation of hADScs into both SMCs and cryoinjured SMCs in the IAS muscle, we prepared Dil-stained hADScs for stem cell tracking. The hADScs (1 × 10^6^ cells/mL) were suspended in serum-free medium prior to injection, mixed with Dil dye (5 μL/mL; Cell Labelling Solution, V-22885, Invitrogen, Waltham, MA, USA), and incubated for 5 min at 37 °C. The labeled cells were centrifuged at 1500 rpm for 5 min. Next, the supernatant was removed and the cells were gently resuspended in complete medium. The wash procedure was repeated twice. The Dil-stained hADScs (1 × 10^6^ cells per site) were injected at the IAS injury site using a 25-gauge needle immediately after cryoinjury. The injection was performed at a depth of less than 1 mm at the intersphincter dissection surface, which was immediately followed by wound closure.

Rats were randomly assigned to the Dil-stained hADSc-treated group (*n* = 10), sham cryoinjury group (*n* = 10), and the normal group (*n* = 10). The sham cryoinjury group underwent intersphincteric dissection and cryoinjury without cell implantation. Rat anal sphincters were harvested as whole specimens, from the anal verge to the rectum, one and two weeks after implantation. After cryoinjured muscle layers were confirmed in the IAS ring specimens harvested from the anal sphincter (Figure 2A,C), we identified implanted cells displaying Dil staining-targeted fluorescence among host cells around the injury site, after confirming a lack of signal in normal tissue and cryoinjured tissue without treatment (Appendix A Figure A1A,B).

Rats were housed separately in the animal laboratory under controlled conditions to optimize animal care. Rats were provided with ad libitum access to rodent feed and water under standard laboratory conditions. This animal study was conducted in accordance with the guidelines and with the approval of the Institutional Animal Care and Use Committee of Hallym University (HMC 2021-1-0317-02).

### 2.6. Statistical Analysis

The minimum sample size was estimated for the random allocation of experimental animals to each group to develop the IAS-specific incontinence model and stem cell implantation in the established animal model according to data from our previous systematic review [14]. Statistically significant differences among groups were determined using one- or two-way analysis of variance (ANOVA), followed by Bonferroni’s post hoc test using GraphPad Prism (9.3.1) (GraphPad Software Inc., San Diego, CA, USA). A *p*-value of less than 0.05 was considered statistically significant.

## 3. Results

The *in vitro* differentiation of hADScs into SMCs was confirmed before in vivo cell implantation. After 14 days of cell growth in smooth muscle inductive medium, differentiated cells with SMC features appeared. These cells displayed α-SMA and calponin expression, which was revealed via immunofluorescence staining. Using RT–PCR analysis, we observed that α-SMA, calponin, smoothelin, caldesmon, and SM22α levels were significantly higher in the hADSc-treated group than in the control group (Figure 2).

Upon confirmation of IAS-specific incontinence induced by cryoinjury, impaired muscle layers accompanying the intact submucosa, outer muscular, and serosa layers were identified via H&E and Masson’s trichrome staining in the cryoinjured group (*n* = 5) compared to the normal group (*n* = 5) (Figure 3). The disappearance of α-SMA and calponin-stained connective tissue in the impaired layer during immunofluorescence analysis confirmed cryoinjury targeting the SMCs in the IAS. RT–PCR analysis showed that SMC markers, including, SM22α, calponin, SMMHC, smoothelin, and SDF-1, were significantly decreased in the cryoinjured group compared with the normal group (SM22α, calponin, SMMHC, and smoothelin, *p* < 0.001; SDF-1, *p* < 0.05). CoL1A1 expression appeared significantly altered after cryoinjury.

Cell treatment, which was performed in the IAS-targeting incontinence model, showed that histologic augmentation of muscular layers stained for α-SMA and calponin was present two weeks after cell injection compared to one week after cell injection in the hADSc-treated group (Figure 4). While tracking labeled cells at the injured site, a population of Dil-stained cells was located at the site of augmented smooth muscle, with a diminished cell count one week after cell injection compared to two weeks after cell injection (Figure 5). Among the normal (*n* = 10), sham (only cryoinjury, *n* = 10), and hADSc-treated groups (*n* = 10), only the hADSc-treated group showed significantly increased levels of α-SMA, SM22α, SMMHC, and smoothelin two weeks after implantation compared to one week after implantation (Figure 4D).

## 4. Discussion

Herein, we first developed an IAS-specific incontinence animal model with molecular and histologic changes in SMCs in the IAS. Our incontinence model deliberately targeted SMCs in the IAS and can reduce the bias caused by involving skeletal muscle cells in previous animal models in which anal sphincterotomy was performed at both the IAS and EAS [20,21,22,23]. In the established model, implanted hADScs augmented SMCs at the injured site. Moreover, cryoinjury might represent a physiological approach, similar to that in a previous study [26,27], as compared to a mechanical procedure, such as simple division, segmental resection or crush injury. The RhoA/ROCK (Ras homolog gene family, member A/Rho-associated, coiled-coil containing serine/threonine kinase) pathway has a critical role in determining basal tone development [14]. ROCK binds to the plasma membrane via its ‘split’ PH and C1 domains [28]. In this study, we considered that rapid freezing at −196 °C could influence the intactness of the plasma membrane, which is essential for functional cell survival, without shifting intracellular water to attain an equilibrium, but avoided the substantial loss of muscle volume, while diminishing the lethality associated with an intermediate temperature zone (−15 to −60 °C) [29]. Although this study could not determine whether newly differentiated SMCs originated from Dil-stained hADScs or host cells, we expect that augmentation of SMCs in the injured area may be appropriate for tissue regeneration corresponding to the myogenic properties of the IAS. In addition, we consider that further translational studies are needed to determine whether SMCs newly regenerated by hADSc treatment can produce an IAS basal tone with an intrinsic force similar to that of the host IAS, since our previous systematic review showed a lower association between the molecular mechanisms of IAS and functional outcomes of treatments due to the complexity of the molecular mechanisms for basal tone and IAS relaxation [14].

Restoration of resting pressure is one among the clinical parameters for symptomatic resolution in patients with anal incontinence [24]. Compared to the voluntary contractility of skeletal structures innervated by the pelvic plexus carrying somatic fibers, the autonomic nerve fibers supplying the IAS are distinct from the nerves entering the EAS via the pelvic plexus [30]. As an intrinsic property, the basal tone is maintained in the smooth muscle of the IAS conjoined with the intestinal muscular layer independent of extrinsic nerve stimulation [31]. Therefore, the functional loss of involuntary myogenic tone in the IAS may be critical in determining various etiologies of AI, including traumatic injury, nerve damage, age, or incontinence-related medical conditions of patients. In elderly patients, involuntary leakage is the symptom most associated with deterioration, compared to that in young women seeking care for incontinence, although the characterization of age-related phenotypes of isolated IAS-induced incontinence might be difficult in a clinical setting [32].

However, to investigate the functional recovery of resting pressure or SMCs in the IAS, several animal models representing the pathophysiology of a dysfunctional IAS have been reported [20,21,22]. After applying intramuscular or intravascular injection of mesenchymal stem cells (MSCs) in an animal model, Salcedo et al. [21,22] reported the functional recovery of anal resting pressure but could not detect cell differentiation or implanted MSCs at the injury site under confocal microscopy or in immunofluorescence assays. These results can be explained by the process of homing and the upregulation of trophic cellular mediators released by stem cells. However, this method may be biased by trapped cells in the systemic circulation and later recovery after remote administration of MSCs [33]. In contrast to the indirect approach, a previous study using surgical implantation of ADSc sheets showed improved anal pressure, using fluorescence in situ hybridization to detect regeneration of implanted cells into SMCs [20]. However, donor cells from the recipient were not identified without the demonstration of the mechanism linking the differentiation of SMCs and functional recovery in this model.

Oh et al. [34,35] showed the functional differentiation of injected myoblast cells labeled with PKH-26 fluorescent dye that binds irreversibly to the cell membrane or DAPI-stained cell nuclei. However, the differentiation into SMCs was weak and not sufficiently clear to establish an association with functional improvement [35]. Various methods, including a biosuture covered with ADScs [36]. Local electrical stimulation followed by stem cell injection [37,38], and direct injection of ADScs combined with bulkamid [23], have been applied to escalate the therapeutic effect of injected cells. Cell differentiation leading to the regeneration of new muscle fibers by implanted cells was not observed in previous studies, despite advanced techniques [34,35,36,37,38]. The success of cellular therapy depends on cell migration to the site of interest, subsequent cell differentiation, and replacement of the fibrous tissue [19]. Furthermore, some studies have suggested various alternative strategies [39] and the necessity of their development based on critical pitfalls in the clinical setting of cell therapy for anal sphincter incontinence [40]. In this context, our outcomes may provide an experimental clue to the mechanism of cellular therapy using hADScs for patients with AI. However, above all, the safety and security of hADSc approaches must be proven before preclinical or clinical application to patients with AI, and the clinician should be aware of severe adverse events related to stem cell therapies [41,42].

Our study has several limitations; We did not measure anal pressure, the contractility of muscle strips using electromyography or the long-term effects of treatment. An understanding of the long-term effect of stem cell therapy remains limited, considering the technical difficulties associated with tracking the implanted stem cells based on long-term cell generation in an animal model [14]. Further study is warranted, requiring advanced techniques to enhance cell proliferation, differentiation, and long-term survival. Cell culture for two weeks may be too short to observe complete cell proliferation and differentiation. Furthermore, the pathophysiology of the AI model in animals, especially rats, may differ from that in humans in terms of sphincter defect size and etiology. It remains uncertain whether the functional improvement in resting pressure in the IAS after cell injection leads to symptomatic resolution of incontinence in a rat model, although it mimics AI in humans [43]. Surgical dissection for injury may induce aligned defective endothelial linings with fenestration of injected cells. The very thin layer of the injured site may complicate targeted injection, despite using microscopy, which leads to leakage of injected material. The very thin surgical plane of intersphincteric dissection might unintentionally induce damage to the skeletal muscle layer at the EAS. Moreover, harvesting targeted tissue, including non-injured or normal tissue, in cross-sections may lead to bias in the quantitative analysis of SMCs in the IAS. This study did not exclude the possibility of the differentiation of implanted stem cells into myofibroblasts during cell growth. Above all, our cell implantation method without biomaterials or shields to prevent cell migration, diffusion, and detachment might be vulnerable to an unfavorable cellular environment with the external pressure of sphincter muscles and fecal passage. Morphometric analysis was not performed due to a shear stress by those factors. During cell tracking, we did not record viable differentiated cells sending Dil signals within the injury site. Further studies should be conducted using simultaneous double staining with varied fluorescence intensity for cell tracking and differentiation.

## 5. Conclusions

We first established an IAS-specific AI animal model. This study demonstrated that implanted hADScs restored impaired smooth muscle at the injury site, indicating that the stem cell fate corresponds to SMCs of the IAS. An advanced technique to enhance cell differentiation by reducing bias and vulnerability is warranted to achieve the critical size of the IAS defect in our newly developed IAS-targeted AI model.

## Figures and Tables

**Figure 1 jcm-12-01632-f001:**
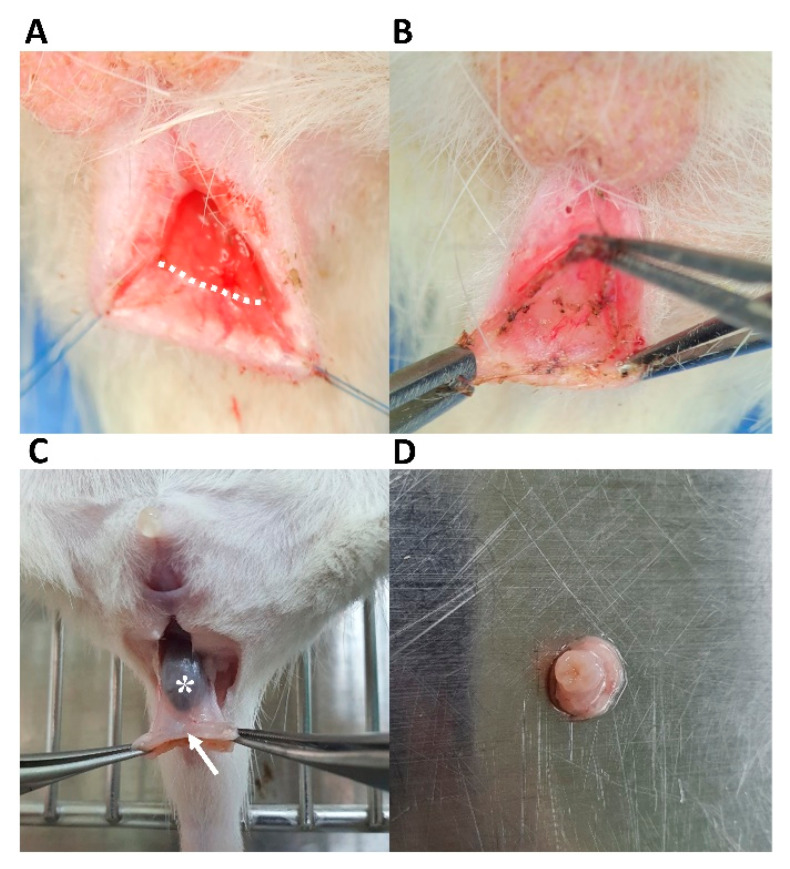
Development of an anal incontinence model targeting the internal anal sphincter. (**A**) A transverse incision (dotted line) was made at the posterior intersphincteric groove. (**B**) Intersphincteric dissection via the space between the submucosal and muscular layers. (**C**) The anal sphincter was harvested using caudal traction at the anoderm (arrow) from the distal rectum with feces *. (**D**) Anal sphincter specimen after removal of the anoderm.

**Figure 2 jcm-12-01632-f002:**
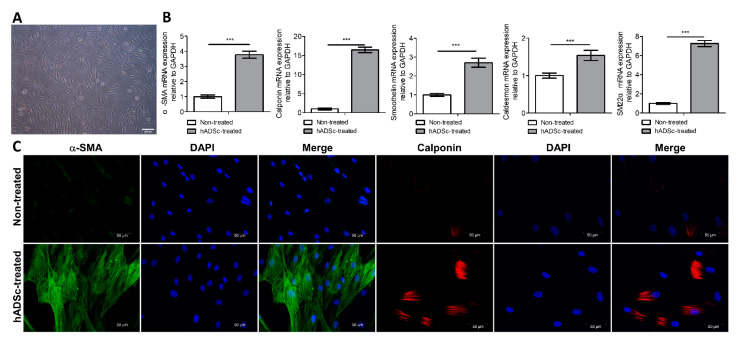
*In vitro* differentiation of hADScs into SMCs. (**A**) Optimal differentiation into SMCs was observed after two weeks when cells were cultured in smooth muscle inductive medium (low-glucose DMEM containing 0.1% gentamicin plus 5 ng/mL TGF-β). 2D-cultured hADScs in uncoated dishes grew to confluence and formed a monolayer of fibroblast-like cells presenting a flat morphology and a stress fiber pattern. (**B**) RT–PCR analysis of *α-SMA*, *calponin*, *smoothelin*, *caldesmon*, and *SM22α* levels between the hADSc-treated group and the control group after cell differentiation. (**C**) Immunofluorescence staining of α-SMA and calponin expression after cell differentiation. SMC, smooth muscle cell; TGF-β, transforming growth factor β; hADSc, human adipose-derived stem cell; RT-PCR, real-time polymerase chain reaction; α-SMA, α-smooth muscle actin; GAPDH, glyceraldehyde 3-phosphate dehydrogenase. Data are expressed as the mean  ±  SD; *** *p*  <  0.001.

**Figure 3 jcm-12-01632-f003:**
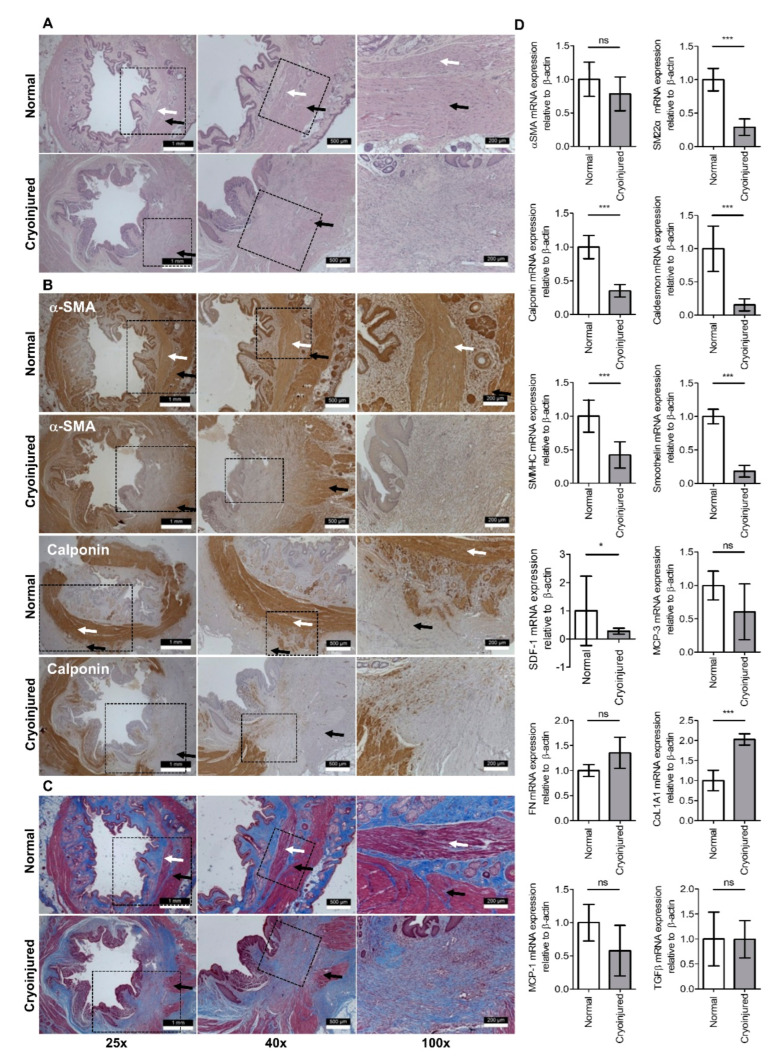
Confirmation of the internal anal sphincter-specific incontinence model. Histopathology showing cryoinjury of smooth muscle in the internal anal sphincter (white arrow) with intact mucosa, submucosa, and circular external sphincter muscle layers (black arrow) compared to the normal anal sphincter on (**A**) H&E staining, (**B**) immunofluorescence staining for α-SMA and calponin, and (**C**) Masson’s trichrome staining. (**D**) RT−PCR analyses of *α−SMA*, *SM22α*, *calponin*, *caldesmon*, *SMMHC*, *smoothelin*, *SDF−1*, *MCP−3*, *FN*, *CoL1A1*, *MCP−1*, and *TFG−β* expression in the cryoinjury group and normal group. H&E, hematoxylin and eosin; α-SMA, α-smooth muscle actin; RT-PCR, real-time polymerase chain reaction; SMMHC, smooth muscle myosin heavy chain; SDF-1, stromal cell-derived factor 1; MCP, monocyte chemoattractant protein; FN, fibronectin; TGF-β, transforming growth factor β. Data are expressed as the mean  ±  SD; * *p*  <  0.05; *** *p*  <  0.001; ns = no statistical significance.

**Figure 4 jcm-12-01632-f004:**
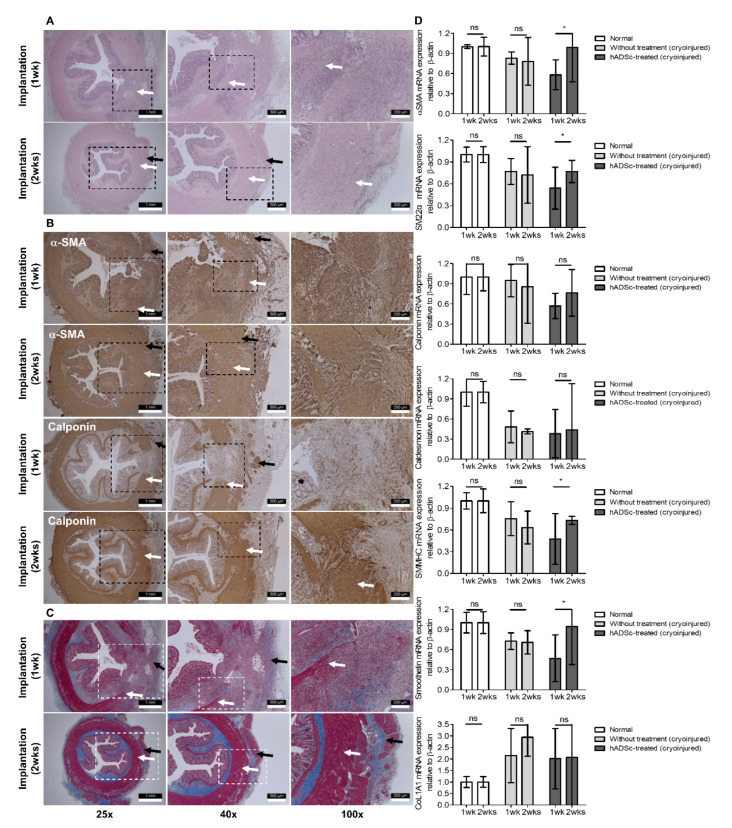
Cell treatment based on the internal anal sphincter-specific incontinence model. Histologic examination after (**A**) H&E staining, (**B**) immunofluorescence staining for α-SMA and calponin, and (**C**) Masson’s trichrome staining 1 and 2 weeks after hADSc implantation at the cryoinjury site (dotted line box). Histology showed the regenerated smooth muscle in the internal anal sphincter (white arrow) with intact mucosa, submucosa, and circular external anal sphincter muscle layers (black arrow). (**D**) RT-PCR analysis of SMC markers in the normal, without treatment (only cryoinjured), and hADSc-treated groups based on *α-SMA*, *SM22α*, *calponin*, *caldesmon*, *SMMHC*, *smoothelin*, and *CoL1A1*. H&E, hematoxylin and eosin; α-SMA, α-smooth muscle actin; hADSc, human adipose-derived stem cell; RT-PCR, real-time polymerase chain reaction; SMMHC, smooth muscle myosin heavy chain. Data are expressed as the mean  ±  SD; * *p*  <  0.05; ns = no statistical significance.

**Figure 5 jcm-12-01632-f005:**
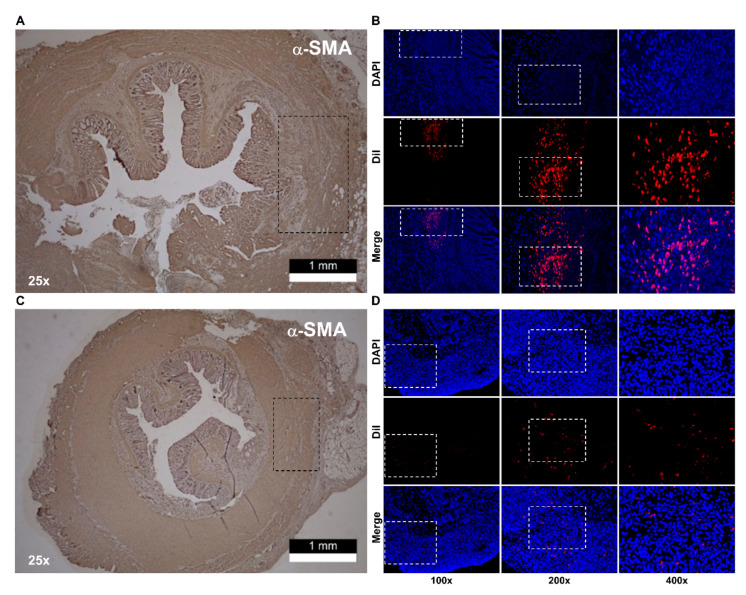
Cell tracking for Dil-labeled hADScs. Histologic examination with fluorescent dye for impaired smooth muscle layer (dotted line box): (**A**) Muscular destruction of α-SMA stained layers in the IAS ring compared to ovoid and longitudinal smooth muscle bundles at non-injury sites, and (**B**) Dil tracking of labeled hADScs with fluorescent dye (dotted line box) one week after treatment. (**C**) Restoration of smooth muscle layer (dotted line box): proliferation of α-SMA stained muscle fibers without infiltration of fibrotic tissue, and (**D**) tracking of labeled hADScs two weeks after treatment (dotted line box). hADSc, human adipose-derived stem cell; α-SMA, α-smooth muscle actin; IAS, internal anal sphincter.

**Table 1 jcm-12-01632-t001:** Primers for smooth muscle cells derived from human adipose stem cells.

Origin	Gene	F/R	Sequence (5′→3′)
Human	α-SMA	F	ACCCAGCACCATGAAGATCA
		R	TTTGCGGTGGACAATGGAAG
Human	Calponin	F	GTGAAGCCCCACGACATTTT
		R	TGATGTTCCGCCCTTCTCTT
Human	Smoothelin	F	GTACGGGCTCAGGAGATTGA
		R	GAAACCTCTGCCTGCTGTTC
Human	Caldesmon	F	AAAACCTACAAAGCCGGCAG
		R	AAACCCCTACCTTCAAGCCA
Human	SM22α	F	TCTTCACTCCTTCCTGCGAG
		R	TATGATCCACTCCACCAGCC
Rat	α-SMA	F	ACTGGGACGACATGGAAAAG
		R	GCCACATACATGGCAGGGACATTG
Rat	SMMHC	F	GATGTGGTGCAGAAAGCTCA
		R	TGAGAATCCATCGGAAAAGG
Rat	Smoothelin	F	TCAAGCAGATGTTGCTGGAC
		R	ACAGAAAGCCATCCCATCAC
Rat	Caldesmon	F	TGGAAGCAGAAGAACAAGAAC
		R	TTCAGCCTCCCTCCTCTC
Rat	SM22α	F	AGGTGTGGCTGAAGAATG
		R	CCTGTTCCATCTGCTTGA
Rat	Calponin	F	ACTTCATGGATGGCCTCAAG
		R	GTGCCAGTTCTGGGTTGACT
Rat	SDF-1	F	CTGAATAGTGGCTCCCAAGGTT
		R	GTGGATCTCGCTCTTCCCTGAC
Rat	MCP-3	F	GCATGGAAGTCTGTGCTGAA
		R	CCGTTCCTACCCCTTAGGAC
Rat	Fibronectin	F	GAAAGGCAACCAGCAGAGTC
		R	CTGGAGTCAAGCCAGACACA
Rat	MCP-1	F	CTATGCAGGTCTCTGTCACGCTTC
		R	CAGCCGACTCATTGGGATCA
Rat	CoL1A1	F	ACAGGCGAACAAGGTGACAGAG
		R	GCCAGGAGAACCAGCAGAGC
Rat	TGF-β	F	AGGGCTACCATGCCAACTTC
		R	CCACGTAGTAGACGATGGGC

α-SMA, α-smooth muscle actin; SMMHC, smooth muscle myosin heavy chain; SDF-1, stromal cell-derived factor 1; MCP, monocyte chemoattractant protein; TGF-β, transforming growth factor β.

## Data Availability

Data are available upon request from the corresponding author.

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
