# Peer review of "Differentiation of Adipose-Derived Stem Cells into Smooth Muscle Cells in an Internal Anal Sphincter-Targeting Anal Incontinence Rat Model"

_jcm, 2023, doi:10.3390/jcm12041632_

Round 1
Reviewer 1 Report
Firstly, I would like to congratulate you by the high quality of the submitted paper. The methodology is excellent, and the information provided has a very high potential clinical relevance.
Its contribution to the international scientific literature will be very important because it is an important and poorly treated healthy problem.
Maybe I would like you to develop more deeply some aspects in your paper. In the following sections, aspects I consider modifiable or revisable of the submitted manuscript will be highlighted.
Literature review is probably a little superficial, 34 references is not very representative for this field of interest.
Related to the TITTLE, I think it could be important to change “animal model” to “rat model”, because authors made the research in a murine model.
In the INTRODUCTION section, we can mention:
· In the line 42 It seems that the last word is lacking, maybe “EAES or nerve injury or damage”?
· In the line 58 it is not clear the meaning of “approval of hADSCs”… Do the authors refer to human use? Do the authors refer to ADSCs original description?
· In the line 59 it seems that authors want to explain that ADSCs have been widely employed in animal and human research dur to their easy isolation method, wide availability, etc… It must be better explained in the sentence.
· In the clinical setting faecal incontinence due to isolated IAS damage is not very frequent I believe. Authors could add a paragraph explaining the real clinical relevance of the damage to IAS.
· I believe that authors must explain deeply that cell therapy has been proven in experimental and clinical settings and that they must mention also another of the available review articles in the field as:
o Gräs S, Tolstrup CK, Lose G. Regenerative medicine provides alternative strategies for the treatment of anal incontinence. Int Urogynecol J. 2017 Mar;28(3):341-350. doi: 10.1007/s00192-016-3064-y. Epub 2016 Jun 16. PMID: 27311602.
o Balaphas A, Meyer J, Meier RPH, Liot E, Buchs NC, Roche B, Toso C, Bühler LH, Gonelle-Gispert C, Ris F. Cell Therapy for Anal Sphincter Incontinence: Where Do We Stand? Cells. 2021 Aug 13;10(8):2086. doi: 10.3390/cells10082086. PMID: 34440855; PMCID: PMC8394955.
· In humans resting anal pressure is not only due to IAS, there is a role of EAS and also of hemorrhoidal tissue.
Talking about the METHODOLOGY:
· Line 167. When the authors explain the cell injections. They must present the material employed (syringe, needle,…) and how it is performed deeply. Previous or later to skin closure?
· In the paper from Trebol et al (reference 31) some bubbles and loss of SCs injected product was observed… Didi t happened also to the authors? How did they deal with this problem?
· What was exactly the plane of the injection? Muscle at both sides of the injury? Injury gap? How was it controlled? Ideally cellular products mut be injected always in the same plane.
· A commentary about anaesthetic methodology must be provided.
· Statistical analysis. What were the employed statistical test to assess if the variables fulfilled criteria of normality? The animal groups have a low N number and authors employed mostly parametric tests.
·
In the RESULTS SECTION:
· Line 232: authors describe findings on experimental groups. Histologic augmentation, … Did they appear in all the animal of each group?
· Concerning presented results and figures about staining, immunofluorescence and PCR. The mentioned findings are homogeneous for all the animals in each experimental group or is there any variability?
· Did the authors performed any morphometric analysis or measure to assess if the observed difference is really relevant or is the effect of selecting the best area?
In the DISCUSION SECTION, there are some aspects to be commented deeply:
· Line 282 to 284: there is a sentence repeated in the text.
· Line 316: moreover to the long-term effect, we do not know if there is long-term survival of implanted or derived from implanted cells…
· Line 323: authors must explain deeply the potential importance of the leakage of the injected material (previously mentioned by Trebol et al as we have mentioned before) and also if the mucosal layer is damaged that could be associated with the faecal contamination of the sphincters described also in the literature.
· Why did the authors select cryoinjury better than a selective incision in IAS? Maybe a surgical section is more controllable than the cryoinjury...
· There are nowadays some papers including nearly 100 human patients treated with stem cells to their faecal incontinence. A paragraph must be dedicated to comment this clinical experience.
· A commentary must be provided to the different cell dosages employed.
· Specific paragraphs must be provided mentioning the supposed mechanism of action in this field and the remaining concerns related to the security of Stem Cells. In this last aspect, in the field of ophthalmology has appeared a clinical severe adverse events potentially relatable to SCs: one paper reported three women suffering from macular degeneration, after undergoing ASC therapies developed complications including vision loss, detached retinas and bleeding and are now totally blind (the ASCs were really SVF) and later another bilateral retinal detachment was reported. [PMID: Pmc5551890 DOI: 10.1056/NEJMoa1609583] [PMID: DOI: 10.3928/23258160-20170829-16
Newly I would like to congratulate authors for their work. Keep working in this way.
Reviewer 2 Report
Dear Authors,
it is a well designed study on rat models of IAS injury.
The methods are clear and well described. The results are good with important message tha, of course, need to be investigated more.
The authors report the limitations of the study correctly.
Just one consideration. It is not discussed that the injection of hADScs just after the cryoinjury may be deeply different from an injection in a chronic IAS defect, as it really happens after delivery or proctologic surgery. In fact, it is usually use for FI not to treat and acute wound but a scar tissue. It might be interesting also to evaluate the regenerative effect on a 3months-anal scar.
It should be reported in the discussion section
Reviewer 3 Report
This is an interesting study on the utilization of adipose stem cells for the treatment of anal incontinence due to internal anal sphincter lesion. While the concepts are truly interesting, the work is disadvantaged by language issues that make the whole manuscript difficult to understand. Moreover, the setting of the experiments are sometimes inappropriate. I would truly like to see a revision version of this work. I think that it is really important to focus on the internal anal sphincter which represent a true challenge in the treatment of anal incontinence.
p.1 l.40. Please use the term 2anal incontinence” which is more accurate. Anal incontinence also encompasses incontinence to gazes.
p.1 l.41-42. Word missing in the sentence. Is it “damage”?
P.2 l. p51-52 I do not think we can speak of heterogeneity or bias. There is simply not study focusing on cellular therapy and IAS. You cannot state that myoblasts have an effect on IAS based only on manometry indirect results of few studies. How does it work? Perhaps this is the result of an increase of tissue stiffness? This observation only gives some clues.
P2 l.58 which approval??
P.4 l.150 which sphincter wound was repaired? IAS? Moreover, the EAS and IAS are sticking together in rats, how do you manage to separate them? What was the learning curve for this experiment? You mention only 10 animals! EAS was also probably damaged. We can see on figure 3 that EAS seem to be interrupted.
P.5 168: The correct control should be cryoinjury and dill. As an alternative you could have created a separated group with only few rats injected with dill labelled ADS, just for cell fate tracking.
.p.5 and p.4 Harvesting after only one or two weeks after injury seems too short. I guess this was a terminal experiment? Animal were euthanized?, please ad details.
Figure2 : regarding expression of alpha-SMA is it really specific to smooth muscle cells?? Myofibroblasts also express alpha-SMA and TGF-beta is a classical trigger for myofibroblast activation. Should be important to compare the level of expression of all those genes with freshly isolated smooth muscle cells form the IAS or another smooth muscle.
p.6 l.210: how the cryoinjury was confirmed? If I understand well, the cells were injected right after injury?
p.6 l.215 RT-PCR of what? Whole sphincters or IAS? How to be selective?
Figure 4: please do not use sham for the cryoinjured group. “Without treatment” should be more appropriate.
Figure4: how to be sure that the increased level of expression of all those genes is not due to myofibroblasts?
p.9 l.264: I’m not sure that your model is only targeting IAS and SMC as some fibers of the EAS were damaged as shown in Fig 3. This model is certainly closer to specific IAS damage than others. The sentence should be modified in order to include this uncertainty.
p.10 l.276: IAS is for sure the main component of anal tonus but I’m not sure one can say that it is the main parameter that can improve patients symptoms. Indeed, while frequently associated, IAS and EAS lesions provide different patterns of anal incontinence.
p.10 l.283 typo
Round 2
Reviewer 1 Report
The new version is much better than the previous one and I believe is enough and satisfies all my previous demands.
The response letter is one of the best I have found before.
Keep working in this way.